# Different Responses to Water Deficit of Two Common Winter Wheat Varieties: Physiological and Biochemical Characteristics

**DOI:** 10.3390/plants12122239

**Published:** 2023-06-07

**Authors:** Antoaneta V. Popova, Gergana Mihailova, Maria Geneva, Violeta Peeva, Elisaveta Kirova, Mariyana Sichanova, Anelia Dobrikova, Katya Georgieva

**Affiliations:** 1Institute of Biophysics and Biomedical Engineering, Bulgarian Academy of Sciences, Acad. Georgi Bonchev Str., Bl. 21, 1113 Sofia, Bulgaria; popova@bio21.bas.bg; 2Institute of Plant Physiology and Genetics, Bulgarian Academy of Sciences, Acad. Georgi Bonchev Str., Bl. 21, 1113 Sofia, Bulgaria; gmihailova@bio21.bas.bg (G.M.); boykova2@yahoo.com (M.G.); vili@bio21.bas.bg (V.P.); elisab@abv.bg (E.K.); msichanova@abv.bg (M.S.); georgieva.katya.m@gmail.com (K.G.)

**Keywords:** dehydration, rehydration, photosynthetic pigments and proteins, primary photosynthetic reactions, chlorophyll fluorescence, antioxidant activity, thermoluminescence

## Abstract

Since water scarcity is one of the main risks for the future of agriculture, studying the ability of different wheat genotypes to tolerate a water deficit is fundamental. This study examined the responses of two hybrid wheat varieties (Gizda and Fermer) with different drought resistance to moderate (3 days) and severe (7 days) drought stress, as well as their post-stress recovery to understand their underlying defense strategies and adaptive mechanisms in more detail. To this end, the dehydration-induced alterations in the electrolyte leakage, photosynthetic pigment content, membrane fluidity, energy interaction between pigment–protein complexes, primary photosynthetic reactions, photosynthetic and stress-induced proteins, and antioxidant responses were analyzed in order to unravel the different physiological and biochemical strategies of both wheat varieties. The results demonstrated that Gizda plants are more tolerant to severe dehydration compared to Fermer, as evidenced by the lower decrease in leaf water and pigment content, lower inhibition of photosystem II (PSII) photochemistry and dissipation of thermal energy, as well as lower dehydrins’ content. Some of defense mechanisms by which Gizda variety can tolerate drought stress involve the maintenance of decreased chlorophyll content in leaves, increased fluidity of the thylakoid membranes causing structural alterations in the photosynthetic apparatus, as well as dehydration-induced accumulation of early light-induced proteins (ELIPs), an increased capacity for PSI cyclic electron transport and enhanced antioxidant enzyme activity (SOD and APX), thus alleviating oxidative damage. Furthermore, the leaf content of total phenols, flavonoids, and lipid-soluble antioxidant metabolites was higher in Gizda than in Fermer.

## 1. Introduction

In the global climate change scenario, drought stress will increase in frequency and severity, thus becoming a major problem for the growth and yield of crop plants, including wheat [1,2,3]. Therefore, exploring the capability of crop species to grow under water deficit conditions is fundamental. The current efforts are focused on the development, evaluation, and study of new crop genotypes with enhanced drought tolerance to increase the world’s food production [4,5,6]. Wheat has played a crucial role in global food security and hence there is a constant need to increase wheat productivity given the continued growth of the world population [7,8,9]. In general, plants respond to drought stress at many levels, as their defense strategies can differ depending on the species, genotype, and developmental stage, as well as the duration and severity of the stress.

The primary effect caused by drought is osmotic stress, whereas the secondary effects are complex and include oxidative stress causing changes in membrane structure and function, damage of membrane lipids, proteins and nucleic acids, and metabolic dysfunction [10], which negatively affects crop development and yield [11,12]. One of the most dangerous consequences of drought stress is the enhanced production of reactive oxygen species (ROS) in different cellular compartments, leading to photooxidative damage causing extensive cellular damage and photosynthetic inhibition [13,14,15]. Furthermore, stress-generated ROS can serve as a signal that triggers plant defense responses by specific signal pathways, the accumulation of protective substances, activation of plant enzyme antioxidant system, etc. [16,17,18].

In addition, it is well established that drought stress initially induces a significant decrease in the chlorophyll, carotenoid, phenols, and relative water content of leaves [19,20], whereas the electrolyte leakage, osmolyte accumulation, antioxidant enzyme activity, and oxidative stress markers increased progressively with drought severity [21,22,23]. Photosynthesis is the main driving force for plant growth and crop grain yield, but the functional state and activity of the photosynthetic apparatus largely depend on the water availability in plants [24,25]. For realizing an effective photosynthetic process, the coordinated functioning of the components of the photosynthetic apparatus is required. Therefore, exposure to different environmental conditions including dehydration can lead to translocation and rearrangement of main pigment-protein complexes which are enabled by the high fluidity of the lipid component of thylakoid membranes [26,27]. It has been also reported that the net photosynthetic rate and stomatal conductance were significantly decreased in different crop cultivars under drought conditions, resulting in an inhibition of photosynthetic processes, thus affecting plants’ productivity [19,20,21,22,23,24,25,26,27,28].

The selection of wheat genotypes is a process related to the creation of new varieties with optimal combinations of valuable biological and economic properties, as well as environmental interactions to improve their resistance to osmotic stress [7,11,25,29]. Moreover, breeding for improved photosynthesis in crop plants is a manageable and useful addition to genetic engineering to enhance crop potential [4,30]. Wheat is a main crop worldwide and is often exposed to a water deficit, especially during seed germination and the phase of crop establishment in early autumn [25]. Therefore, the assessment of drought tolerance at the seedling stage is the most important trait for screening drought tolerance in wheat breeding, since it affects all subsequent stages and ultimately the grain yield [31,32].

The Pulse-Amplitude-Modulated (PAM) chlorophyll fluorescence is a widely applied technique for the evaluation of photosynthetic function and monitoring of stress responses [33,34,35,36], since the acclimation of plant photosynthetic processes to environmental changes is a key component of plant adaptation to abiotic stress [37]. Moreover, the PSII photochemistry is the most sensitive indicator of early drought stress and thus could be used for the selection of drought-tolerant plants [15,38]. The chlorophyll fluorescence, which relates to the photosynthetic activity, electron flow, and chlorophyll concentration, can be used to identify the resistance of wheat genotypes to drought stress [24,29,39]. It has been found [39] that the chlorophyll fluorescence parameter F_v_/F_m_ is positively correlated with the yield of different wheat genotypes under optimal and drought conditions. The photochemical process under water stress is also characterized by a decrease in photosynthetic-related proteins [40]. The dehydration process is accompanied by the accumulation of different stress-induced proteins such as dehydrins, ELIPs, and HSPs [41,42,43]. Dehydrins are hydrophilic proteins belonging to group 2 of LEA proteins. They have protective functions stabilizing proteins and membranes during stress [44]. ELIPs, a part of the light-harvesting chlorophyll *a*/*b*-binding proteins’ superfamily, have protective functions against photooxidative damage in thylakoid membranes [45].

Our recent study [46] revealed, with fast biochemical stress markers, that the new variety, Gizda, demonstrates a higher tolerance to the water deficit for 7 days and better post-stress recovery in comparison to the older variety, Fermer. However, the molecular mechanisms underlying their different responses to the water deficit are still unclear. The aim of the present study was to unravel the molecular mechanisms involved in the responses of both common wheat varieties (*Triticum aestivum* L., Gizda and Fermer) at the early developmental stage to moderate (3 days) and severe (7 days) drought stress, as well as their recovery capacity. To this end, we followed the changes in their electrolyte leakage, photosynthetic pigment content, primary photosynthetic reactions, energy interaction between pigment–protein complexes, and photosynthetic and antioxidant enzyme activities to identify the different physiological strategies between resistant and sensitive wheat varieties. Understanding the mechanisms by which plants respond and adapt to drought stress is crucial for the development of stress-resistant crop plants, which is important for ensuring global crop availability.

## 2. Results

### 2.1. Relative Water Content (RWC) and Electrolyte Leakage (EL)

RWC of leaves from the Gizda variety slightly decreased under moderate water stress (3 days of water deprivation) but it was not statistically significant, whereas RWC of Fermer was reduced by 40% (*p* ≤ 0.05; Figure 1a). Severe water stress (7 days) resulted in a strong reduction in RWC of both varieties by about 70%. While RWC of Gizda was completely recovered after 3 days of rehydration, it was still about 20% lower than the control in Fermer (*p* ≤ 0.05).

Changes in EL during the dehydration of the two wheat varieties were followed as a measure of membrane integrity and the extent of stress injury (Figure 1b). EL increased significantly only under severe water stress conditions when RWC was reduced by about 70%. Regardless of the similar RWC, EL increased 5- and 15-fold in Gizda and Fermer, respectively. The fast recovery of EL after 3 days of rehydration indicated that the observed changes were reversible.

### 2.2. Alterations in the Photosynthetic Pigment Content

The changes in the content of chlorophylls (Chl *a* and Chl *b*) and total carotenoids (Car) in leaves of dehydrated plants from both wheat varieties expressed as a percentage from the respective controls (100% hydrated plants of the same age as the dehydrated ones) are presented in Figure 2. The leaf pigment content of Fermer was higher than that of Gizda under control conditions (Chl *a* = 14.81 and 11.49, Chl *b* = 4.89 and 2.70, and Car = 5.59 and 4.24 mg g^–1^ DW, respectively). The pigment analysis of leaves from both varieties subjected to dehydration revealed that during the first 3 days of the water deficit, only slight changes were observed in the total chlorophyll (Chl *a* and Chl *b*) and Car content, whereas after 7 days of dehydration (severe stress), a stronger reduction occurred, which was more pronounced in Fermer, by about 30 and 40% for Chl *a* and Chl *b*, respectively (Figure 2a,b). Both varieties recovered the chlorophyll content after rewatering for 3 days, but only in Gizda was it completely restored. Furthermore, in the control plants, the Chl *a*/*b* ratio for Gizda was 4.09, while for Fermer it was significantly lower: 3.23 (Appendix A). The dehydration-induced changes in the Chl *a*/*b* ratio were observed in both wheat varieties after the application of severe drought stress but were fully recovered after the rehydration in Gizda only.

The alterations in total Car content followed the same trend of changes as the Chl *a* content for both varieties, but there were no statistically significant differences between them except for the 3rd day of dehydration (Figure 2c).

### 2.3. Photochemical Activity of PSII and PSI during Dehydration and after Rehydration

Data on the allocation of absorbed light energy under dehydration is presented in Figure 3. The main part of the excitation energy in the control and moderately dehydrated plants from both varieties was used for photochemistry. The maximum quantum efficiency of the PSII electron transport, Y(II), decreased by 10% after 3 days of drought in Gizda and Fermer, but the most significant differences between the two varieties were observed after 7 days. The values of ΦPSII were reduced by 37% and 66% in Gizda and Fermer, respectively. During dehydration, the proportion of light energy dissipated as the light-induced non-photochemical fluorescence quenching, Y(NPQ), and non-regulated heat dissipation, Y(NO), increased. In severely desiccated leaves, the values of Y(NPQ) increased more than two- and three-fold in plants from Gizda and Fermer, respectively, whereas those of Y(NO) were 50% and 100% higher compared to the control. The photochemical activity of PSII was completely recovered following rehydration.

The PSI activity was assessed by measuring the extent of the far-red (FR) light-induced absorbance change at 820 nm (∆A_820–860_), which reflects the oxidation of P_700_ to P_700_^+^ (Figure 4). The photochemical activity of PSI was not significantly affected under moderate drought stress. In contrast to PSII, the PSI efficiency increased slightly in Gizda and more significantly in Fermer (*p* ≤ 0.05) in severely dehydrated (7 days) leaves. In fact, an almost 70% reduction in PSII activity in the latter was accompanied by a 40% enhancement in PSI activity.

When the FR light was turned off, the kinetics of the subsequent P_700_^+^ decay in the dark are presumed to primarily reflect the rates of cyclic electron transport around PSI [47,48]. The rates of P_700_^+^ re-reduction, *t*_1/2_, slightly declined in Gizda and increased by 20% in Fermer as a result of drought stress (Figure 4b). In general, the rate of cyclic electron transport around PSI was faster in Gizda, compared to Fermer.

### 2.4. Thermoluminescence

Data presented in Figure 5 are results from experiments carried out with two flashes (2F) excitation measurements after a 3–4 h dark adaptation period. The 2F for control leaves produced a maximal overall emission and contained a B band generated by S_2/3_Q_B_^–^ peaking at a temperature of 26–27 °C, S_2/3_ being the states of the oxygen-evolving complex storing two and three positive charges; Q_B_ being the secondary quinone acceptor. The sharper band around 46 °C (S_2/3_Q_B_+e), which appeared at the upper edge of the signal, is the so-called afterglow band (AG), which depends on the back transfer of electrons from stroma to the acceptor side of PSII. The two varieties demonstrated similar properties of the B and AG bands in control leaves (Figure 5a,b). The temperature of the B band dropped significantly by 10 °C in Fermer, while only a small non-statistically significant decrease was noticed in Gizda during dehydration (Figure 5c). Therefore, the intensity of the B band of Gizda demonstrated stability to drought stress, indicating that the primary photochemical reactions remain resistant. In addition, the B band area drastically decreased in Fermer under severe drought (Figure 5d), and the dehydration resulted in an almost complete disappearance of the AG band, unlike in Gizda plants (Figure 5a,b). Moreover, drought stress induced an increase in the AG intensity of dehydrated Gizda leaves; however, this was not statistically significant. The rehydration partly restored the suppressed B band in Fermer and caused a rise in the AG band above the control level in both varieties.

### 2.5. Fluidity of Lipid Phase of Thylakoid Membranes as Affected by Water Deficit

The membrane fluidity of thylakoid membranes isolated from leaves of both wheat varieties (Gizda and Fermer) was assessed by the degree of fluorescence polarization (P) of the hydrophobic probe DPH that is evenly distributed between stacked and stroma-exposed thylakoids. Data about the membrane fluidity of Gizda and Fermer thylakoids after water deprivation for 3 and 7 days, and after the rehydration of plants are shown in Figure 6. The calculated value for P of thylakoid membranes from well-watered (control) plants was 0.184 for Gizda, while for Fermer plants it was much higher, at 0.219, suggesting that the fluidity of the lipid phase of Gizda thylakoids was higher in comparison with that of Fermer. During the whole experimental setup, the membrane lipid order in thylakoids of Gizda and Fermer did not demonstrate significant alterations, with the only exception being the thylakoids of recovered Fermer plants that showed a statistically significant decrease in fluidity.

### 2.6. Alterations in Energy Interaction between Pigment-Protein Complexes after Dehydration and Following Recovery

The effect of dehydration for 3 and 7 days and after rehydration of Gizda and Fermer plants on energy transfer between main pigment–protein complexes was assessed in isolated thylakoid membranes by analyzing chlorophyll fluorescence emission spectra at low temperature (77 K) and excitation by 436 nm. In the emission spectra at 77 K, two main peaks are resolved at 685 and 735 nm, and are emitted by the reaction center of the PSII and PSI complex, respectively, and a shoulder at 695 nm originating from the inner antenna of PSII (CP47) [49,50]. In Figure 7, the fluorescence ratios F685/F695 (a) and F735/F685 (b) that characterize the energy interaction in the PSII complex and energy transfer between PSII and PSI complexes are presented, respectively. The calculated fluorescence ratio F685/F695 in thylakoids of Fermer was lower for the whole experimental setup in comparison with that of Gizda. This ratio of F735/F685 in control plants was very similar for Gizda and Fermer. However, the dehydration led to a gradual time-dependent increase of F735/F685. For Gizda plants, the ratio of F735/F685 was elevated from 1.215 in control thylakoids to 1.288 and 1.415 after 3 and 7 days of dehydration, respectively, which represent an increase of 6 and 16%, respectively. The dehydration-induced rise in F735/F685 was much stronger expressed for Fermer thylakoids—from 1.231 in the control thylakoids to 1.541 and 1.704 for the respective time of dehydration (3 and 7 days) that corresponded to a 25 and 38% increase. After rewatering for 3 days, the F735/F685 ratio for the Gizda variety remained the same as after 7 days of dehydration, while for Fermer it decreased significantly after rewatering.

### 2.7. Protein Abundance during Dehydration and after Rehydration

#### 2.7.1. Photosynthetic Proteins

**Figure 8 plants-12-02239-f008:**
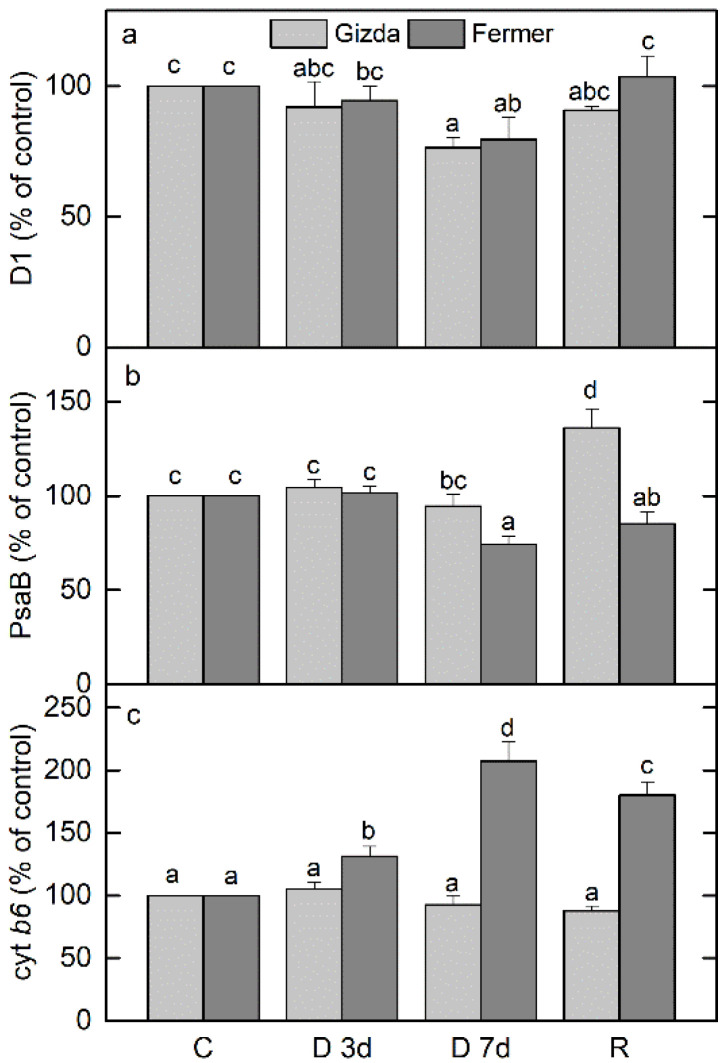
Changes in the content of the main thylakoid-related proteins D1 (**a**), PsaB (**b**) and cyt *b*_6_ (**c**) of Gizda and Fermer varieties in control (C), dehydrated for 3 days (D 3d) or 7 days (D 7d), and rehydrated (R) plants. The abundances of proteins are presented in percentage of the control values (C). Values are given as mean ± SE. The same letters within a graph indicate no significant differences assessed by Fisher’s LSD test (*p* ≤ 0.05) after performing ANOVA.

The protein changes during the dehydration as well as after the rehydration of both wheat varieties, Gizda and Fermer, were studied by Western blot using specific antibodies raised against D1, PsaB, and cyt *b6* (Figure 8). The content of PSII reaction center protein D1 decreased by only 8% and 6% after 3 days, and 24% and 20% after 7 days of dehydration (*p* ≤ 0.05) of Gizda and Fermer, respectively. The rehydration of plants fully recovered the amount of D1 in Fermer, while in Gizda its content reached 91% of control values. PSI reaction center protein PsaB abundance was not significantly affected after 3 days of dehydration in both varieties; however, its content declined by 26% after 7 days under drought only in Fermer leaves (*p* ≤ 0.05). Rehydration increased the level of PsaB up to 145% in Gizda, while in Fermer the protein content did not recover to control levels (85%). Accumulation of cyt *b*_6_ under drought was detected in Fermer leaves up to 207%, and during the recovery of plants, the protein abundance remained high (180%). Interestingly, the cyt *b*_6_ amount was not significantly affected during drought stress and after the rehydration of Gizda plants (93% and 88%, respectively).

#### 2.7.2. Protective Proteins

Specific antibodies raised against ELIP proteins and the K-segment of dehydrins were used to study the effect of dehydration on the amount of protective proteins in the two wheat varieties. Western blot signals demonstrated that dehydrins were not present in control leaves throughout the experiment. Seven days of dehydration led to an enormous accumulation of dehydrins in Fermer leaves (Figure 9). The molecular weight of the detected protein ranged between 60 to 13 kDa. Three major bands could be distinguished—around 22, 28, and 60 kDa. Faint dehydrin signals could be observed 3 days after dehydration as well after the rehydration of Fermer plants. No dehydrin signals were detected in the leaves of the Gizda variety during the experiment.

The protein pattern of ELIPs of both wheat varieties Gizda and Fermer are presented in Figure 9. Three isoforms of ELIPs, major and minor bands, could be detected in wheat plants with a molecular weight between 23 and 20 kDa. ELIPs were constitutively expressed in controls of both varieties. Dehydration increased the ELIP content, especially in Gizda plants. The recovery after rehydration demonstrated a decline in the ELIP abundance.

### 2.8. Antioxidant Power

The activities of the antioxidant enzymes in the leaves of Gizda and Fermer plants were determined on the 3rd and 7th day of water deficiency and after rewatering (Figure 10).

The SOD and GPX activities were significantly higher in the well-watered Fermer plants in comparison with Gizda plants. The imposition of moderate water stress led to an increase in SOD, CAT, and APX activity in Gizda leaves, while on the 7th day of drought stress, the antioxidant enzyme activities decreased, and only the activity of SOD and APX was maintained at a level higher than in the control (Figure 10). In Fermer, a reduction in SOD, CAT and GPX activity was recorded as early as the 3rd day of drought and was further reduced on the 7th day. Regarding the APX activity, the dehydration caused an increase in the enzyme activity of Gizda leaves to the same extent on the 3rd and the 7th day of the dehydration. At the same time in Fermer leaves, the APX activity increased on the 3rd day, followed by a decline on the 7th day compared to the control plants. In Gizda leaves, there was no significant change in the activity of GPX regardless of the duration of drought stress. The results obtained for the levels of the enzyme activities with antioxidant power in Gizda leaves after 3 days of rehydration demonstrated that SOD, CAT, APX, and GPX were with lower levels in comparison to the control well-watered plants. However, in the leaves of Fermer plants, the activity of enzymes SOD, CAT, and GPX increased after rehydration but was still lower in comparison with control plants.

Higher content of total phenols, flavonoids, and lipid-soluble antioxidant metabolites (LS-AOM) was measured in the leaves of Gizda compared with those of Fermer under well-watered and drought-stressed conditions (Figure 11). Total phenols, flavonoids, water-soluble antioxidant metabolites (WS-AOM), and total antioxidant activity measured by the ferric reducing antioxidant power (FRAP) method decreased in Gizda leaves on the 3rd day of drought stress and retained the lower value on the 7th day. Only DPPH radical scavenging activity increased on the 3rd day of drought with a subsequent reduction on the 7th day. Regarding the response of Fermer to water stress, a decrease in the WS-AOM content, as well as in the antioxidant capacity assayed by two methods (DPPH radical scavenging activity and FRAP) was observed on the 3rd and 7th day. The application of the 3 day-rehydration enhanced the WS-AOM and FRAP in Gizda leaves but the values remained lower than the control plants. The dehydration did not cause changes in the content of flavonoids and LS-AOM in the Fermer leaves.

## 3. Discussion

### 3.1. Drought-Induced Changes in RWC and EL 

Plants are often exposed to drought stress at the seedling stage; therefore, studying the resistance of different wheat genotypes at the seedling stage could help in the selection of drought-tolerant cultivars [31,32]. Drought-tolerant plants have less reduction in water content, higher membrane stability, and photosynthetic activity. RWC is a measure of plant water status, reflecting the metabolic activity in tissues and plants’ capacity for osmotic adjustment [51], and is used as an index for dehydration tolerance [20,25,52]. Its stronger reduction in Fermer plants under moderate stress indicates lower drought tolerance in comparison to Gizda (Figure 1a). In addition, the higher sensitivity of Fermer was confirmed as well by the stronger destabilization of cell membranes under a severe water deficit, as determined by the extent of EL (Figure 1b).

### 3.2. Drought-Induced Changes in Photosynthetic Apparatus

#### 3.2.1. Pigments

The leaf chlorophyll content is an important factor for plant productivity since it is directly proportional to the photosynthetic rate and plant growth [19,53]. The decrease in Chl content under drought stress conditions is a characteristic indicator for oxidative stress and could be the result of chlorophyll photooxidation and/or degradation [6,20,21]. In addition, carotenoids in higher plants have many roles, including the regulation of light harvesting [54], ROS-scavenging, and preventing oxidative damage, thereby increasing plant stress tolerance [55]. The decline in the abundance of photosynthetic pigments could have a negative impact on the effective photosynthetic performance and crop yield [51,56]. A higher Chl and Car content under stress conditions has also been associated with the tolerance of plants, as the decrease in pigment content correlates with decreased RWC [20], which was demonstrated by our results indicating a higher tolerance of Gizda than Fermer. Previous studies with different wheat and maize genotypes have demonstrated that under drought stress the leaf Chl and Car content in drought-tolerant cultivars is higher compared to sensitive cultivars [6,20]. Our data revealed that Fermer is a more sensitive wheat variety, which sensed a stronger water deficit and did not restore the drought-induced decrease in leaf pigment content after rewatering, while the Gizda variety completely recovered leaf pigment content after rewatering (Figure 2), which suggests its better tolerance to drought in comparison to the Fermer variety. Moreover, the observed increase in Chl *a*/*b* ratio for Fermer plants under severe drought stress, due to the stronger reduction of Chl *b* content (Figure 2 and Appendix A), indicates a decrease in the light-harvesting antenna of PSII and reduced stacking degree of thylakoid membranes [57,58]. A similar increase in the Chl *a*/*b* ratio has been reported in different wheat cultivars under water-deficient conditions [20]. In addition, the observed lower Chl content and higher Chl *a*/*b* ratio in control Gizda plants in comparison with Fermer plants might be one of the reasons for the greater tolerance of Gizda to stress conditions, as it has been suggested that the reduced antenna size could better protect PSII from photooxidative stress, reducing the ROS generation and dissipation of thermal energy by NPQ [58].

#### 3.2.2. Photochemical Activity of PSII and PSI

Chlorophyll fluorescence measurements are widely used for monitoring the effect of drought stress on photosynthetic activity [59]. They can provide useful information on excitation and energy transfer, primary photochemistry, and the operating quantum efficiency of the electron transport through PSII [60]. It has been demonstrated that fluorescence parameters can be used for selecting cultivars more resistant to water stress [61,62]. The stronger reduction of Y(II) under severe water stress conditions in Fermer compared to Gizda confirmed its higher sensitivity. The drought caused damage to PSII, resulting in significant decreases in the quantum yield of PSII, which was accompanied by a decreased amount of photosynthetic pigments, and D1 protein, as well as an enhanced quantum yield of light-induced non-photochemical fluorescence quenching Y(NPQ), and a quantum yield of non-regulated heat dissipation, Y(NO). It should be noted that Y(NPQ) plays a major protective role under moderate and especially under severe water loss. Indeed, energy dissipation was higher in Fermer, in which photochemical activity was more inhibited. On the other hand, the reduction in PSII activity with increasing the extent of water loss was accompanied by increased PSI activity and it was higher in the more sensitive variety, Fermer. A similar drought-induced increase in PSI activity has also been previously observed in wheat cultivars [25], as this could be a defense mechanism to protect PSI reaction centers from photoinhibition by P700 oxidation. Three days of rehydration were enough for the complete recovery of PSII activity in both wheat varieties. Thus, the inactivation of PSII was reversible due to the activation of drought defense mechanisms, while Y(NPQ) and PSI activity were still higher than the control in Fermer.

The abundance of photosynthetic proteins D1 and PsaB demonstrated a decrease after 7 days of dehydration in Fermer, while in Gizda, only D1 content was reduced. Interestingly, during the rehydration of Gizda plants, PsaB was strongly accumulated, while cyt *b*_6_ abundance was significantly increased during dehydration and after recovery only in Fermer plants. Decreased levels of D1 and D2 proteins are typical for plants under drought, and this is associated with the downregulation of photochemical reactions [40,63]. Our results demonstrated that Fermer is more susceptible to drought stress. Previously, the downregulation of genes encoding PSI core proteins PsaA and PsaB under water stress suggested a lower drought tolerance of maize [64] and rice plants [58].

#### 3.2.3. Thermoluminescence

Thermoluminescence, observed in a 0–60 °C range, provides information on the stabilization of separated by light charge pairs in the PSII [65], assimilatory potential, and cyclic electron transfer capacity [66]. The thermoluminescence flash excitation experiments carried out with leaves of dark-adapted control plants, Gizda and Fermer, revealed fully functional PSII and produced a B band, whose intensity reached a maximum after two flashes (2F). The other band that was observed at around 46 °C was assigned to AG emission that could also appear in healthy leaves after xenon flash illumination. The dehydration of leaves at water shortage influenced photochemical reactions and thermoluminescence revealed different modifications of the obtained B and AG bands. The temperature downshift of the B band during the dehydration stress detected in Fermer is an indication of the destabilization of charges S_2/3_Q_B_^−^, whereas the B band remained resistant during dehydration and rehydration in Gizda. In the parallel to B band temperature maximum (T_max_) that decreased with the progressive dehydration of Fermer plants, the intensity of the band also declined (Figure 5b–d). This parameter is a measure of active PSII centers. The destroyed PSII centers as a result of the photoinhibition that usually occurred in such conditions probably contributed to the non-photochemical quenching. The increase in the AG band intensity in dehydrated Gizda leaves after excitation by 2F can be explained either by the observed higher capacity for cyclic electron transfer pathways (Figure 4b) or a higher NADPH ^+^ ATP potential, that favors the electron transfer from the PSI acceptor pool to the PSII quinone acceptor pool following cyclic pathways [67,68,69]. Under severe drought stress, a decline of the AG band intensity was observed in Fermer. The rehydration caused a rise in the AG band above the control level in both cultivars, as already observed for other wheat cultivars [68]. Moreover, AG T_max_ was found to shift to lower temperatures in Gizda (D 7d and R), which indicates that the cyclic pathways are activated and are complementary to the measured, faster P_700_^+^ re-reduction after FR light.

#### 3.2.4. Fluidity of Lipid Phase of Thylakoid Membranes

The lipid matrix is an essential component of thylakoid membranes, characterized by a high degree of unsaturation of the fatty acyl chains of membrane lipids. The lipid component of thylakoid membranes comprises four main lipid classes; nearly 80% of them are represented by the galactolipids monogalactosyldiacylglycerol (MGDG) and digalactosyl-diacylglycerol (DGDG), which contain three and more double bonds per fatty chain [14,70]. This high degree of unsaturation provides a fluid environment for the structural and functional integrity and reorganization of the main photosynthetic complexes, especially from exposure to environmental stress conditions such as strong light, extreme temperatures, and salt stress [70,71,72]. An alteration of pigment–protein complexes and the formation of the inverted hexagonal phase has been reported after dehydration of *Craterostigma pumilum* [26]. Our results indicate that the fluidity of the lipid matrix, as estimated by the anisotropy of the fluorescent probe DPH, is higher in the control Gizda thylakoid membranes in comparison with that of Fermer membranes at control conditions. This is an indication that the abundance of unsaturated lipid species is higher in Gizda, thus facilitating the structural reorganization and translocation of photosynthetic complexes. In this way, the structural flexibility required for the adaptation of the photosynthetic apparatus to changes in environmental conditions is better provided. The dehydration of plants of both varieties for a period up to 7 days did not lead to statistically significant alterations in the fluidity of thylakoid membranes’ lipid bilayer.

#### 3.2.5. Energy Interaction between the Main Pigment–Protein Complexes

The effects of severe dehydration and rewatering on energy interaction and transfer between photosynthetic complexes were estimated in both wheat varieties, Gizda and Fermer, by recording and analyzing the emission fluorescence spectra at 77 K. The calculated ratio F685/F695 in thylakoid membranes from control plants was much lower in Fermer thylakoids than in Gizda. This is an indication that either the connection between the inner antenna of PSII, CP47, and the reaction center of the PSII complex is not very effective and/or the transfer of energy between these complexes is less efficient in Fermer thylakoid membranes. Under the water deficit, the energy interaction within the PSII complex was not altered in Fermer, while in Gizda, some significant interruption of energy transfer between CP47 and the reaction center of PSII was detected that was successfully restored after the rewatering of plants. The dehydration-induced alterations in Gizda and Fermer thylakoid membranes were more pronounced in respect to an energy transfer from the complex of PSII to PSI, as evidenced by the ratio F735/F685. For thylakoids of well-watered plants of both varieties, there was no statistically significant difference in F735/F685, suggesting a comparable efficiency of energy delivery to the PSI complex. With the onset of water deficiency, the energy transfer to PSI complex was gradually increased in a time-dependent manner that was expressed more strongly for thylakoids of Fermer. This increase of F735/F685 can be due to a number of reasons including the higher population of PSI complexes and/or to translocation of some PSII-LHCII complexes to the stroma-exposed PSI complexes [73,74]. The observed elevation of F735/F685 in the Fermer thylakoids was significantly restored on the reset of watering, while for the Gizda thylakoids, no restoration was detected. As the ratio F735/F685 in thylakoids from the well-watered plants was comparable for both varieties, we suggest that the observed dehydration-induced increase in the ratio F735/F685 is more likely due to the unstacking of thylakoid membranes that was more significant in Fermer thylakoids, which is also confirmed by the observed increase in the Chl *a*/*b* ratio under severe stress. The results from 77 K fluorescence emission spectra (ratios F685/F695) also indicated that in Fermer thylakoids, the energy transfer from the inner antenna (CP47) to the reaction center of PSII (D1) is less effective in comparison with Gizda.

#### 3.2.6. Stress-Induced Proteins

Dehydrins act as molecular chaperones and ROS scavengers in plant cells during stress [75]. Western blot results demonstrated that dehydrins were absent in control plants and they accumulate only during dehydration in the leaves of Fermer plants where three major bands could be distinguished. The lack of dehydrin transcripts in different unstressed wheat plants was reported by Rampino et al. [11]. The accumulation of dehydrins’ transcripts in winter wheat and barley plants in response to drought stress and low temperatures was previously observed [76,77]. Our results confirmed the essential role of dehydrins to maintain and stabilize the subcellular milieu during drought. Gizda plants did not accumulate dehydrins in their cells during water deprivation, although they are more tolerant to drought compared to Fermer plants. Many authors demonstrated that the response to drought stress is genotype-specific [78].

Unlike dehydrins, we detected ELIP proteins by immunoblot analysis in all thylakoid samples investigated for both varieties. The photooxidative damage of chloroplasts is a result of the imbalance of energy supply and utilization during dehydration via carbohydrate metabolism [79]. ELIPs stabilize photosynthetic complexes and enhance thermal energy dissipation in thylakoid membranes [80,81]. A higher abundance of ELIPs in the process of dehydration correlated with the higher non-photochemical quenching, elevated cyclic electron transport and decreased temperature of the B band of both wheat cultivars. Enhanced ELIP transcript levels in wheat plants under different stress conditions were reported [82]. Moreover, the accumulation of ELIPs in thylakoids correlates with the inactivation of PSII and decreased levels of D1 and photosynthetic pigments [83].

### 3.3. Effect of Water Stress Deficits on the Antioxidant Power

Comparing the changes in the activity of antioxidant enzymes in studied wheat varieties indicated that at short drought stress the activity of SOD, CAT, APX and GPX significantly increased in Gizda, but on the 7th day of the drought, their levels were reduced. A reduction of the SOD, CAT and GPX enzyme activities was registered as early as the third day of dehydration in Fermer. Higher values of the content of total phenols, flavonoids, and LS-AOM were detected in Gizda leaves compared with that of Fermer at well-watered and drought conditions, which may also account for its better stress tolerance.

Plants often grow in severe environmental conditions, and due to abiotic and biotic stress, their growth is inhibited. In order to be more persistent against the harmful effects of reactive oxygen species (ROS) over-accumulated under drought conditions, plants unlock an enzymatic and non-enzymatic mechanism to scavenge ROS. The balance between ROS and the antioxidant system is crucial for plant survival and adaptation to stress. It is generally indicated that the SOD, CAT, and POD enzymes have a big impact on how wheat genotypes respond to drought stress. In addition to having a variety-specific dependence between drought and the level of enzymatic and non-enzymatic antioxidant activity, the duration of the drought and wheat development stage is also important. A lower level of lipid peroxidation and hydrogen peroxide content in the Gizda leaves than in the Fermer wheat variety was reported [46]. Therefore, to prevent cellular electrolyte leakage, the integrity of the lipid membrane is maintained during drought stress in Gizda plants. These results corresponded with the higher level of the activity of the major enzymes, CAT and APX, responding to the scavenging H_2_O_2_ produced during drought stress. Hence, to prevent the plant from oxidative damage, more efficient antioxidative defense mechanisms act in Gizda than in Fermer, and maintains the stress-generated O_2_^•−^ and H_2_O_2_ at an optimal level. The same results of minimal accumulations of O_2_^•−^ and H_2_O_2_ under drought conditions, accompanied by minimum lipid peroxidation and electrolyte leakage, have been obtained for drought-tolerant maize genotypes [6] and *S. persica* [18]. According to Abid et al. [84], during the first 5 days of drought, wheat plants demonstrated a significant increase in the activity of SOD, the enzyme that converts O_2_^•−^ into H_2_O_2_, followed by CAT and APX activities that reduce H_2_O_2_ to water to prevent oxidation damage. The tolerant Gizda variety demonstrated enhanced antioxidant enzyme activities (SOD, APX, and GPX) and a lower accumulation of MDA, which is a sign of the increased detoxification of ROS. The sensitive Fermer variety demonstrated less ability to increase and maintain antioxidant enzyme activities under drought stress, leading to a worse recovery than the more tolerant variety. At a longer 7-day drought, the tissues have become extremely sensitive to ROS attacks, leading to a decrease in their levels of activity. Bano et al. [85] reported that in the tolerant wheat genotype (cv. Chakwal-97), imposed at drought stress at the tillering stage for a period of three days, a significant increase in SOD activity and no marked increase in POD activity was recorded. After rehydration, SOD activity has retained its high levels, while a significant decrease in POD activity was recorded. In the drought-susceptible genotype (cv. Punjab-96), the SOD and POD activity were not significantly influenced by drought stress, but, after rewatering, the activities of these enzymes were increased. The CAT activity has been enhanced in five studied drought-tolerant wheat genotypes, while in sensitive wheat varieties, a reduction was detected [86]. Although the total content of phenols, flavonoids, and lipid-soluble metabolites was higher in Gizda leaves in comparison with Fermer, the level of radical scavenging activity in Gizda was lower than that in Farmer. This can be explained by the fact that the radical scavenging activity level is not determined only by the content of phenols and flavonoids. The antioxidant capacity is assayed by the reduction in the odd electron of a nitrogen atom in the stable free radical DPPH (1,1-diphenyl-2-picryl-hydrazyl) by receiving a hydrogen atom from all hydrogen donor compounds to DPPH-H [87]. Therefore, in complex biological systems, there are other antioxidant compounds besides phenols and flavonoids that are responsible for the reduction of free radicals [88]. Considering the changes in enzymatic and non-enzymatic antioxidant activity, it can be assumed that the variety Gizda is more tolerant than Fermer to drought stress.

## 4. Materials and Methods

### 4.1. Plant Material and Drought Conditions

Experiments were performed with two varieties of Bulgarian common winter wheat (*Triticum aestivum* L., Gizda and Fermer) from the Breeding program of the Institute of Plant Genetic Resources, Sadovo, Bulgaria, obtained by hybridization [46]. Both varieties are mutant lines of the variety Pobeda. Fermer was obtained by the treatment of dry seeds with gamma rays of 50 Gy; Gizda was obtained by the treatment of dry seeds with 1 mM of NaH_3_ [44]. Seeds (30 per pot) were sown in pots (d = 15.5 cm, h = 14.5 cm) filled with soil from the region of Sadovo, Bulgaria, containing 11.4% humus, 486 mg/kg K_2_O, 654 mg/kg P_2_O_5_, and 265 mg/kg total N (pH 7.6), as described in [44], and grown in a climate chamber FytoScope FS-RI 1600 (Photon Systems Instruments, Drasov, Czech Republic) under controlled conditions for 21 days. The light intensity during the whole setup was 250 ± 10 μmol photons m^−2^ s^−1^, the day/night photoperiod was 16/8 h, the day/night temperature was 20/18 °C, and the relative humidity was 60%. When the third leaf was developed, pots were divided into 2 groups. Plants in 9 pots were well-watered regularly (every day) and served as control (100% irrigated). The watering of plants in the second set of 15 pots was stopped for 3 days and 7 days to follow the extent of dehydration of the seedlings. After 7 days of water deprivation, the plants were rewatered for 3 days (R) to evaluate their ability to recover after dehydration stress. Samples were collected parallelly from control and dehydrated plants (at the same age) on the 3rd and 7th day of dehydration and after the recovery period of 3 days. The middle part of the second fully developed leaf was used for the registration of chlorophyll a fluorescence and thermoluminescence. For the isolation of thylakoid membranes, developed leaves for every respective group of plants were used. 

### 4.2. Determination of RWC

The RWC of leaves was determined gravimetrically by weighing them before and after oven-drying at 80 °C to a constant mass and expressed as a percentage of water content in dehydrated tissue compared to water-saturated tissues, using the equation:RWC (%) = (FW − DW)/(TW − DW) × 100
where FW—fresh weight, DW—dry weight, and TW—turgid weight. TW was measured on leaves maintained for 12–16 h at 4 °C in the dark, floating on water.

### 4.3. Electrolyte Leakage

Electrolyte leakage from leaf tissues (0.1:10 ratio of leaf tissue to dH_2_O) was measured with a conductivity meter (EC 215, Hanna Instruments, Woonsocket, RI, USA). After a 24 h incubation of leaf disks in double-distilled water on an orbital shaker (OS-20, Boeco & CO GmbH, Hamburg, Germany), the conductivity (μS cm^−1^) of the solution was measured. The maximum leakage of the tissue was determined after boiling the leaves for 15 min at 100 °C. The results are expressed as a percentage of the maximum leakage.

### 4.4. Photosynthetic Pigments’ Content

Photosynthetic pigments were extracted from leaf material with ice-cold 80% acetone (*v*/*v*) in dim light, as described in Gerganova et al. [89]. Leaf material (40 mg) was grinded by a mortar and pestle at 4 °C in dim light and the extracts were centrifuged in sealed tubes at 4500× *g* and 4 °C for 15 min. The photosynthetic pigment content—chlorophyll a (Chl *a*), chlorophyll b (Chl *b*), and carotenoids (Car)—was determined spectrophotometrically (UV-VIS Specord 210 Plus, Analytic Jena, Jena, Germany) in the clear supernatant following the method of Lichtenthaler [90]. Samples were collected from the second leaf of different wheat plants on the 3rd and 7th day after ending the watering and after 3 days of recovery. At every time point, four parallel samples were collected from normally watered and treated plants. Mean values ± SE (*n* = 4) were calculated and expressed on a dry weight basis (mg pigment g^−1^ DW).

### 4.5. Chlorophyll a Fluorescence Induction

Chl *a* fluorescence induction was measured with a portable fluorometer PAM-2500 (Heinz Walz GmbH, Effeltrich, Germany). The leaves were dark-adapted for 15 min and a PAR of 90 μmol (photon) m^−2^ s^−1^ was used for the measurements. All of the used basic parameters were given by PamWin-3 software (version 3.05, Heinz Walz GmbH, Effeltrich, Germany). The maximum efficiency of the PSII photochemistry was calculated as F_v_/F_m_ immediately after the pre-darkening period. The actual efficiency of PSII electron transport during illumination was estimated at a steady state as ΦPSII = (F_m_′ − F_s_)/F_m_′ [91]. The quantum yield of the light-induced non-photochemical fluorescence quenching was calculated as (Y)NPQ = (F_m_/F_m_′) − 1 and the quantum yield of non-regulated heat dissipation and fluorescence emission as Y(NO) = F_s_/F_m_ [92]. All three yield parameters sum up to 1: Y(II) + Y(NPQ) + Y(NO) = 1.

### 4.6. P700 Measurements

The redox state of P700 was monitored in vivo as ΔA820 nm absorption changes. A Walz ED 700DW-E emitter/detector unit was connected to a PAM 101E main control unit (Heinz Walz GmbH, Effeltrich, Germany). P_700_ was oxidized by far-red (FR) light from a photodiode (FR-102, Heinz Walz GmbH, Effeltrich, Germany). The intensity of FR light was 13.4 W m^−2^. FR light was controlled by the PAM 102 unit and applied via the multibranched fiber optic system.

### 4.7. Thermoluminescence Measurements

Thermoluminescence emission from leaves of dark-adapted plants was measured with a home-made apparatus described in detail in Zeinalov and Maslenkova [93]. Freshly excised leaf pieces from the middle part of the leaf from dark-adapted plants were placed immediately on the sample holder aluminum surface at 20 °C and covered with a plexiglass window. After cooling by liquid nitrogen to 1 °C and maintained for one minute, the samples were illuminated with saturating (4J) single turnover xenon flashes (10 µs half-band, 1 Hz frequency) of white light. Then, the sample was warmed up at a 0.5 °C s^−1^ heating rate. The temperature of the sample was measured with a tiny thermocouple, inserted in the sample holder. The luminescence was detected by R943-02, Hamamatsu Photonics. Thermoluminescence signals from data files were smoothed and the temperature maximum (T_max_) and area of the individual bands were determined after signal decomposition by using Origin Lab 8.5.

### 4.8. Isolation of Thylakoid Membranes

Thylakoid membranes were isolated from leaves of control (plants watered every day and the same age as the treated ones), dehydrated for 3 or 7 days, and after rewatering (recovered) plants, as described by Harrison and Melis [94]. Briefly, around 10 g of leaf material was homogenized with a 100 mL isolation buffer containing 0.4 M of sucrose, 5 mM of MgCl_2_, 10 mM of NaCl, and 50 mM of TRICINE (pH 7.8). The homogenate was filtered through two layers of Miracloth, and the filtrate was centrifuged for 5 min at 4500× *g*. The pellet was resuspended in 50 mL isolation buffer that did not contain sucrose. After 5 min incubation, a second centrifugation was performed for 10 min at 6000× *g*. The final pellet was resuspended in the isolation buffer and used for further assays. The whole procedure was performed under dim light and at 4 °C. The pigment content in thylakoid membranes was determined in 80% acetone extracts following the method and equations of Lichtenthaler [90].

### 4.9. 77 K Chlorophyll Fluorescence Measurements

The energy transfer and interaction between the main pigment–protein complexes were evaluated by recording and analyzing fluorescence emission spectra of thylakoid membranes at a low temperature (77 K). Isolated thylakoid membranes were resuspended in a medium containing 0.33 M of sucrose, 5 mM of MgCl_2_, 10 mM of NaCl, and 20 mM of Tricine (pH 7.5) at a concentration of 10 µg of Chl ml^−1^. Every sample was transferred to a quartz tube for fluorescence measurement and immediately frozen in liquid nitrogen. Fluorescence emission spectra were recorded by a spectrofluorometer (Jobin Yvon JY3, Division d’Instruments S.A., Longjumeau, France) equipped with a red-sensitive photomultiplier (Hamamatsu R928, Hamamatsu Photonics, Hamamatsu City, Japan) and a low-temperature device. The width of the slits was 4 nm. Emission spectra were recorded at excitation with 436 nm (excitation of Chl *a*). At every time point of the experiment, four parallel samples were recorded and analyzed. The spectra were analyzed by Origin 7.0 (Microcal Software) after baseline correction.

### 4.10. Steady-State Fluorescence Polarization of DPH

The fluidity of the lipid phase of isolated thylakoid membranes was assessed by evaluation of the degree of polarization (P) of the fluorescence emitted from the probe 1,6-diphenyl-1,3,5-hexatriene (DPH) at room temperature, as described previously [74,95]. The hydrophobic fluorescent probe DPH is evenly distributed between all lipid domains and no energy transfer occurs between DPH and photosynthetic pigments. The probe was added to a final concentration of 2.5 µM from a stock solution to thylakoid membranes (5 µg Chl ml^−1^), resuspended in buffer (0.33 M of sucrose, 5 mM of MgCl_2_, 10 mM of NaCl, and 20 mM of Tricine, with a pH of 7.5). Measurements were performed at room temperature using a fluorometer JASCO FP8300 (Jasco, Tokyo, Japan), equipped with polarization filters. Fluorescence was excited at 360 nm and registered at 450 nm. The slit widths were 10 nm. The degree of polarization (P) was estimated using the formula described previously [95].

### 4.11. Protein Isolation, SDS-PAGE, and Western Blot

Isolated thylakoid samples were solubilized in the sample buffer (50 mM of Tris-HCl, pH of 6.8, 2% (*w*/*v*) SDS, 2% (*v*/*v*) *β*-mercaptoethanol, and 10% (*v*/*v*) glycerol). Samples were separated on SDS-PAGE (SE260 Mighty Small II, Hoefer, Holliston, MA, USA) according to Laemmli [96], modified by adding 8% glycerol to stacking, and separating gels using a constant current of 20 mA per gel. Each lane contains thylakoid samples corresponding to 1.5 μg Chl. Using semi-dry transfer (TE70X, Hoefer, Holliston, MA, USA), the proteins were blotted on nitrocellulose membrane for 90 min at a current of 1 mA cm^−2^. ROTI^®^Mark TRICOLOR (Carl Roth GmbH + Co. KG, Karlsruhe, Germany) was used as a pre-stained protein standard for monitoring electrophoretic separation and transfer efficiency. Blots were probed with primary antibodies against PsbA (D1, AS05 084), PsaB (AS10 695), PetB (cyt *b6*, AS18 4169), ELIPs (AS06 147A), and dehydrin K-segment (AS07 206A) (Agrisera, Vännäs, Sweden). Horseradish peroxidase-conjugated goat anti-rabbit secondary antibody was used (AS09 602, Agrisera, Vännäs, Sweden). The resulting bands were visualized by chemiluminescence, and signals were recorded on X-ray Blue films (Carestream Dental LLC, Atlanta, GA, USA). Films were scanned using an Epson Perfection V850 PRO scanner, and the densitometry was made by Gel-Pro Analyzer software (version 4.0, Media Cybernetic, Rockville, MD, USA).

### 4.12. Antioxidant Activity Analysis

For the enzyme extraction, 0.5 g leaf fresh samples were homogenized with a mortar and pestle in a cold (0–4 °C) 0.1 mM K-phosphate buffer with a pH of 7.8, containing 2.0 mM of Na-EDTA (ethylenediaminetetraacetic acid), 1 mM of PMSF (phenylmethylsulfonyl fluoride), 2% polyvinylpyrrolidone K-40 (*w*/*v*), and 10% (*v*/*v*) glycerol. The homogenate was centrifuged at 12,000× *g* for 30 min and the supernatant was used as a crude enzyme extract [97]. The total SOD (EC 1.15.1.1) activity was measured by its ability to inhibit the photochemical reduction of nitroblue tetrazolium (NBT) [98]. The blue formazane produced by the NBT photo-reduction was measured spectrophotometrically (Shimadzu UV-1601, Tokyo, Japan) as an increase in absorbance at 560 nm. One unit of SOD activity was defined as the amount (mg) of protein required to cause a 50% inhibition of the NBT photoreduction rate. The CAT (EC 1.11.1.6) activity was measured by adding 100 μL of enzymatic extract to 3 mL of solution containing 50 mM of potassium phosphate buffer with a pH of 7.0, 0.1 M of EDTA, and 15 mM of H_2_O_2_ [99]. The decrease in absorbance at 240 nm was monitored. The results were expressed as units—mol H_2_O_2_ destroyed per min per mg of protein. The APX (EC 1.11.1.1) activity was assayed by adding 0.2 mL of crude enzyme extract to the reaction mixture containing 50 mM of phosphate buffer with a pH of 7.0, 0.1 M EDTA, and 0.5 mM of ascorbate and 0.1 mM of H_2_O_2_ [100]. The ascorbate oxidation was followed spectrophotometrically at 290 nm for 1 min. The enzyme activity was calculated using the molar extinction coefficient for ascorbate (2.8 mM^−1^ cm^−1^), and the results were expressed as µmol of H_2_O_2_ destroyed per min per mg of protein. Leaf samples’ extract (0.1 mL) used for enzyme GPX (EC 1.11.1.7) analyses was added to a reaction mixture (1 mL) containing 0.1 mM of phosphate buffer with a pH of 7.0, 0.1 μM of EDTA, 5.0 mM of guaiacol, and 15.0 mM of H_2_O_2_ [101]. GPX activity was defined by the amount of tetraguaiacol formed using its molar extinction coefficient (26.6 mM^−1^ cm^−1^). The results were expressed as nmol H_2_O_2_ mg^−1^ protein min^−1^. Soluble protein content was determined by Bradford [102] using bovine serum albumin as a standard.

For the analyses of metabolites with antioxidant capacity, dry plant samples (0.3 g) were ground and extracted with 80% (*v*/*v*) methanol. The total phenolic compounds (TPC) were determined by the Folin–Ciocalteu colorimetric method using caffeic acid as an equivalent [103]. The total flavonoid content in the leaves was measured spectrophotometrically using the standard curve of catechin [104]. The DPPH radical scavenging activity utilizes the ability of antioxidants to reduce the artificial stable free radical DPPH• (1,1-diphenyl-2-picrylhydrazyl), resulting in a reduction in the blue color. To estimate the radical scavenging capacity, a decrease in absorbance at 517 nm was detected [105]. The ferric-reducing antioxidant power (FRAP) was measured by using the ability to reduce ferric ions to the ferrous by a reductant at low pH [106]. Spectrophotometric quantification of water-soluble (WS-AOM) and lipid-soluble (LS-AOM) metabolites with antioxidant capacity, expressed as equivalents of ascorbate and α-tocopherol, were performed through the formation of the phospho-molybdenum complex [107]. The method has been characterized by linearity interval, repeatability and reproducibility, and molar absorption coefficients for the quantitation of WS- and LS-AOM expressed as equivalents of ascorbate and α-tocopherol. Absorption coefficients were: (3.4 ± 0.1) × 103 M^−1^ cm^−1^ for ascorbic acid and (4.0 ± 0.1) × 103 M^−1^ cm^−1^ for α-tocopherol.

### 4.13. Statistics

Data were presented as mean values (±SE). Mean values were calculated from 4 parallel samples for each time point. Changes in the investigated parameters between wheat varieties were statistically compared by the Fisher’s least significant difference test at *p* ≤ 0.05 following ANOVA. A statistical software package (Statgraphics Plus, version 5.1 for Windows, The Plains, VA, USA) was used.

## 5. Conclusions

The responses of two wheat varieties at the seedling stage to drought stress demonstrated that they use different protective mechanisms to survive unfavorable conditions. The presented data suggest that the Gizda variety is more tolerant to severe dehydration stress compared to Fermer, as revealed by the lower reduction in water content, photosynthetic pigments, active PSII centers, and D1 proteins, as well as higher cell membrane stability and photosynthetic activity, and lower dehydrins’ content. Some of the defense strategies of the Gizda variety to maintain a stable photosynthetic performance under drought stress conditions involve decreased chlorophyll content in control leaves, the accumulation of early light-induced proteins (ELIPs), increased capacity for PSI cyclic electron transport, enhanced antioxidant enzyme activity (SOD, APX, and GPX), and content of non-enzymatic antioxidants (phenols, flavonoids, and lipid-soluble metabolites), which alleviate the oxidative damage. Furthermore, the increased fluidity of the lipid matrix of the thylakoid membrane from Gizda plants facilitates the structural reorganization and translocation of the photosynthetic complexes in the photosynthetic apparatus and also serves as a defense mechanism. Gizda plants also demonstrated better recovery in comparison to Fermer after three days of rewatering.

## Figures and Tables

**Figure 1 plants-12-02239-f001:**
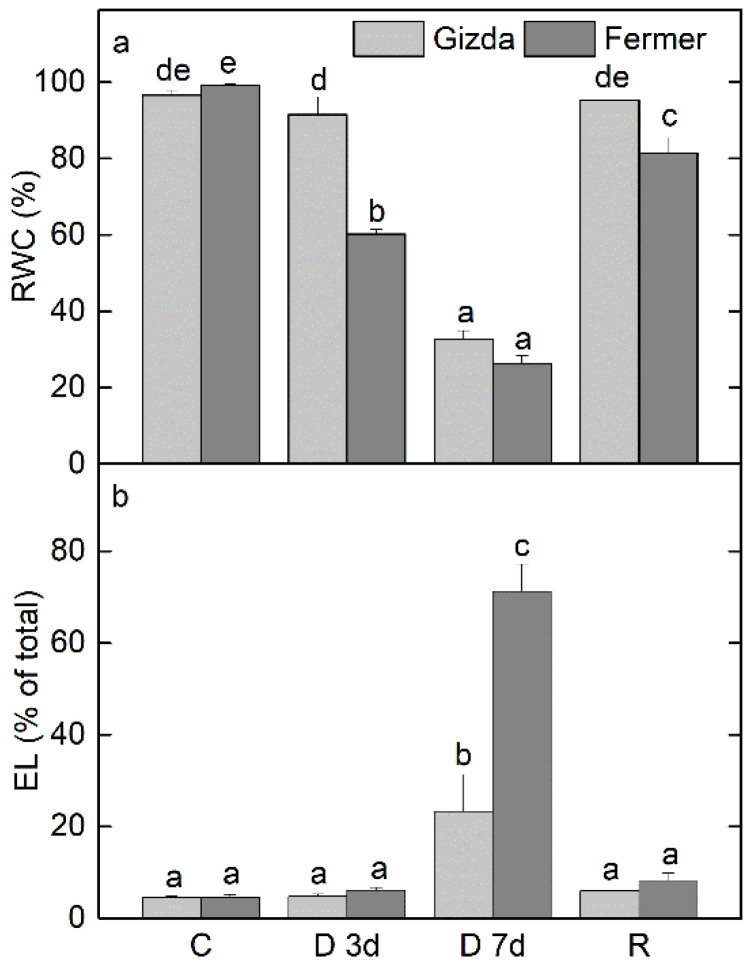
RWC (**a**) and EL (**b**) of leaves of control, well-hydrated plants (C), and after 3 days (D 3d) and 7 days (D 7d) of dehydration and 3 days of rehydration (R) of wheat plants (Gizda and Fermer). Values are presented as mean ± SE (*n* = 4). Significant differences between values are indicated by different letters according to Fisher’s LSD test (*p* ≤ 0.05) of multifactor ANOVA analysis.

**Figure 2 plants-12-02239-f002:**
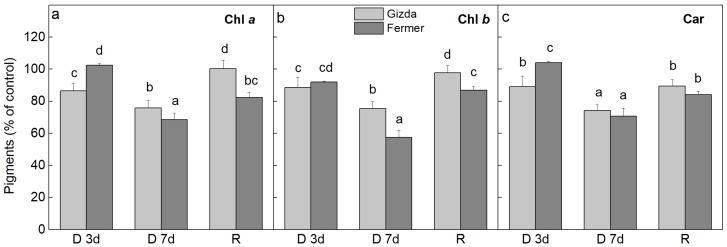
Effect of water deprivation for 3 (D 3d) and 7 (D 7d) days, and after 3 days of rehydration (R) of two wheat varieties (Gizda and Fermer) on photosynthetic pigment content. Results are presented as a percentage from the respective control. Mean values ± SE (*n* = 4) were calculated from four parallel samples at each time point. (**a**) Chl *a*; (**b**) Chl *b*; (**c**) Carotenoids (Car). Different letters indicate significant differences between values at *p* < 0.05 as estimated by Fisher’s LSD test of multifactor ANOVA analysis.

**Figure 3 plants-12-02239-f003:**
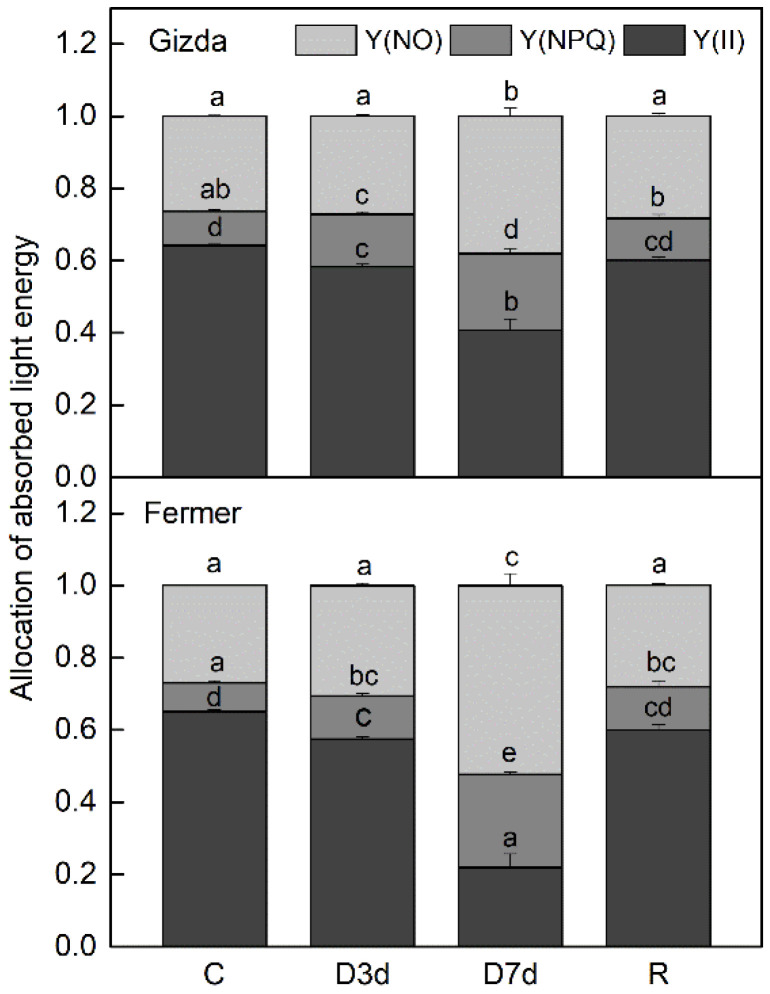
Changes in the quantum yield of PSII electron transport during illumination, Y(II), the quantum yield of light-induced non-photochemical fluorescence quenching, (Y)NPQ, and the quantum yield of non-regulated heat dissipation and fluorescence emission, Y(NO), in control (C), dehydrated for 3 days (D 3d) or 7 days (D 7d) and rehydrated (R) wheat plants (Gizda and Fermer). Values are presented as mean ± SE (*n* = 4). Significant differences between values are indicated by different letters according to Fisher’s LSD test (*p* ≤ 0.05) of multifactor ANOVA analysis.

**Figure 4 plants-12-02239-f004:**
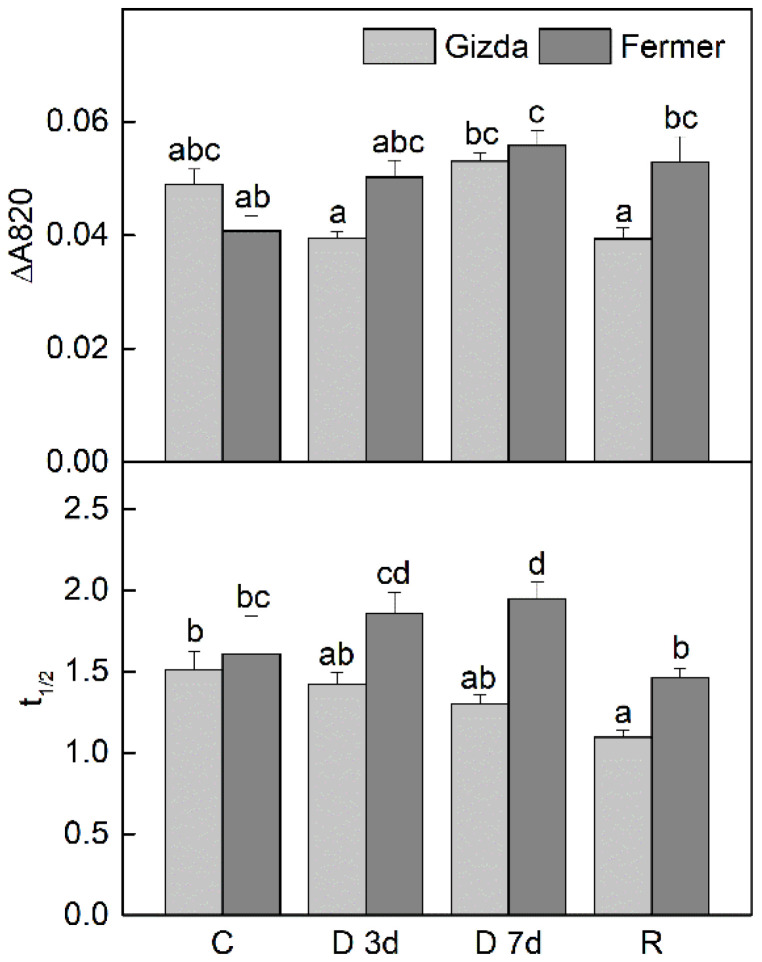
Changes in the levels of FR light-induced P700 photooxidation (P_700_^+^) measured by ΔA_820–860_ and half-times of P_700_^+^ re-reduction (*t*_1/2_) after turning off the FR light illumination in control (C), dehydrated for 3 days (D 3d) or 7 days (D 7d), and rehydrated (R) wheat plants (Gizda and Fermer). Values are presented as mean ± SE (*n* = 4). Significant differences between values are indicated by different letters according to Fisher’s LSD test (*p* ≤ 0.05) of multifactor ANOVA analysis.

**Figure 5 plants-12-02239-f005:**
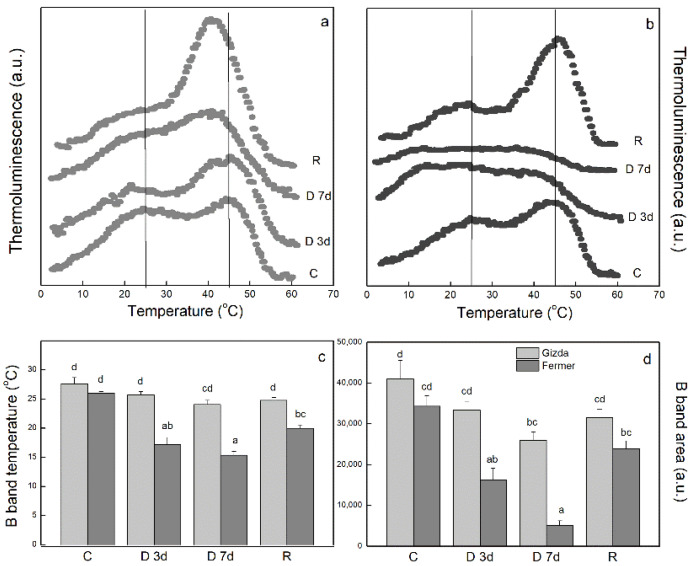
Thermoluminescence curves of leaves from two wheat varieties: Gizda (**a**) and Fermer (**b**), from Control (C), dehydrated for 3 days (D 3d) or 7 days (D 7D), and rehydrated (R) plants. Curves were recorded after excitation by two flashes after at least 3 h of dark adaptation. Values of T_max_ (**c**) and integrated intensity of B band (**d**) obtained after curves decomposition. Means ± SE (*n* = 3). Significant differences between values are indicated by different letters according to Fisher’s LSD test (*p* ≤ 0.05) of multifactor ANOVA analysis.

**Figure 6 plants-12-02239-f006:**
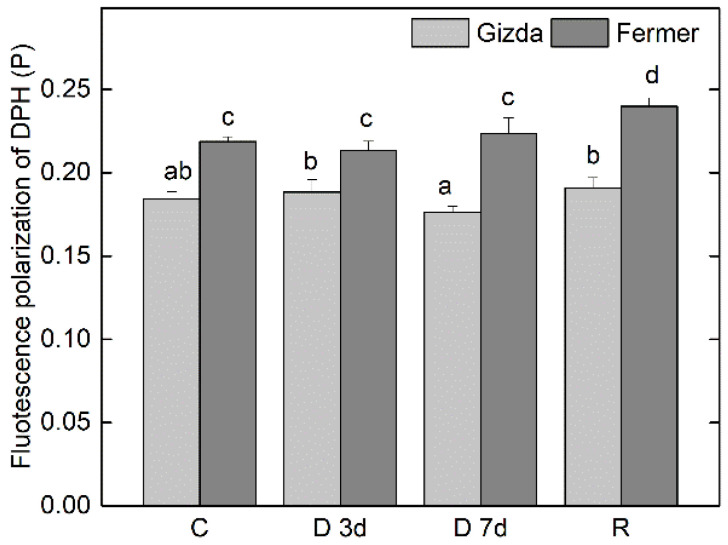
Fluidity of the lipid phase, estimated by the degree of fluorescence polarization of DPH (P) in thylakoid membranes isolated from Gizda and Fermer leaves, affected by dehydration for 3 (D 3d) and 7 days (D 7d) and after rewatering (R) in comparison with control (well-watered) plants (C). Values are presented as mean ± SE (*n* = 4). Significant differences between values are indicated by different letters according to Fisher’s LSD test (*p* ≤ 0.05) of multifactor ANOVA analysis.

**Figure 7 plants-12-02239-f007:**
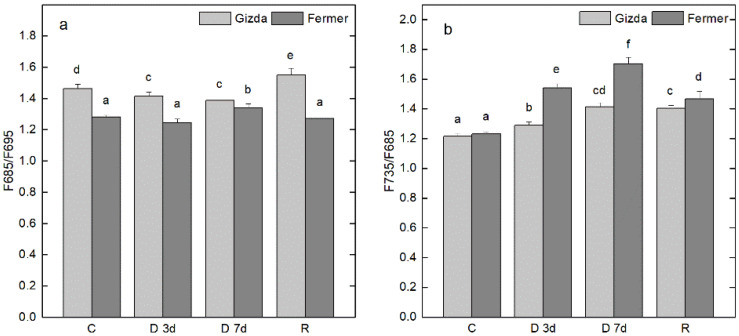
Fluorescence ratios F685/F695 (**a**) and F735/F685 (**b**) in isolated thylakoid membranes from control (well-watered) plants (C), dehydrated for 3 days (D 3d) or 7 days (D 7d) and rewatered (R) wheat plants (Gizda and Fermer). Fluorescence ratios were calculated from 77 K fluorescence emission spectra at excitation with 436 nm after baseline correction. Mean values ± SE (*n* = 4) were calculated from four parallel samples for each time point. Significant differences between values are indicated by different letters according to Fisher’s LSD test (*p* ≤ 0.05) of multifactor ANOVA analysis.

**Figure 9 plants-12-02239-f009:**
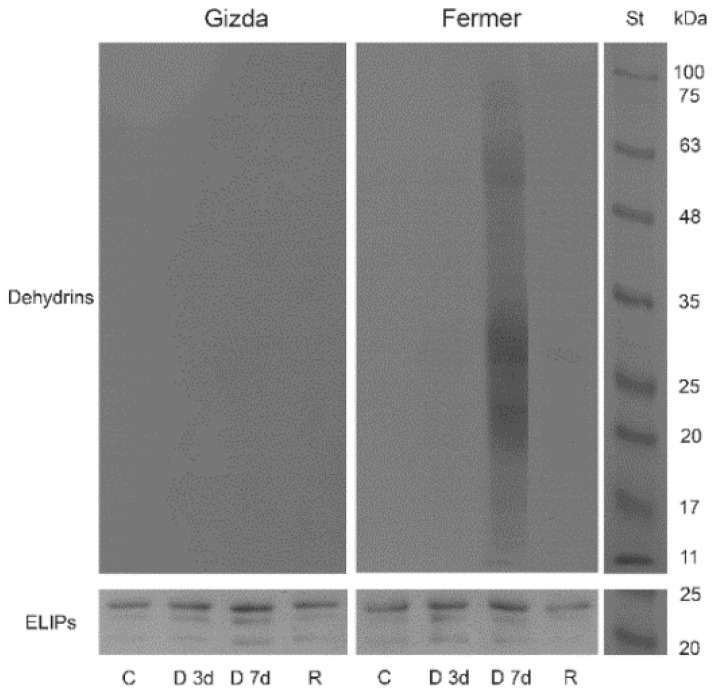
Representative Western blots of dehydrins and ELIPs of Gizda and Fermer varieties in control, normally watered (C), dehydrated for 3 days (D 3d) or 7 days (D 7d), and rehydrated (R) plants. Thylakoid samples corresponding to 1.5 μg Chl were applied per lane. St: ROTI^®^Mark TRICOLOR (Carl Roth GmbH + Co. KG, Karlsruhe, Germany).

**Figure 10 plants-12-02239-f010:**
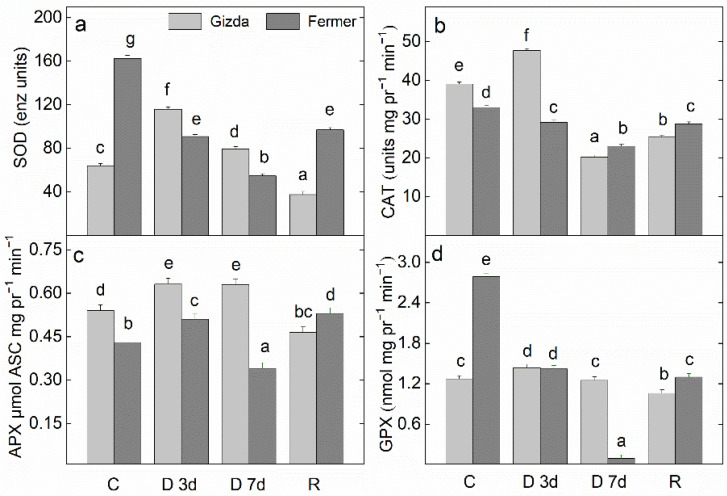
The activity of antioxidant enzymes (**a**) superoxide dismutase (SOD), (**b**) catalase (CAT), (**c**) ascorbate peroxidase (APX), and (**d**) guaiacol peroxidase (GPX) in the leaves of two wheat varieties (Gizda and Fermer): normally watered (C), dehydrated for 3 days (D 3d) or 7 days (D 7d), and rehydrated (R) plants. Values are given as mean ± SE (*n* = 4); different letters indicate significant differences assessed by Fisher LSD test (*p* ≤ 0.05) after performing multifactor ANOVA analysis.

**Figure 11 plants-12-02239-f011:**
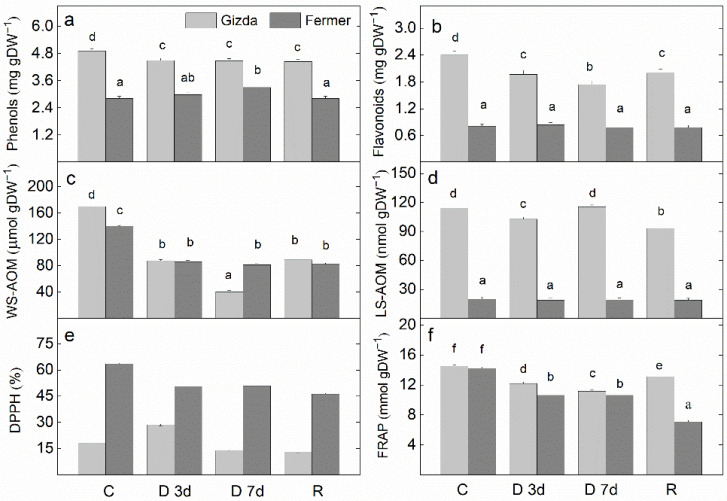
Content of metabolites with antioxidant power and total antioxidant potential in Gizda and Fermer plants: normally watered (C), dehydrated for 3 days (D 3d) or 7 days (D 7d), and rehydrated (R) plants. (**a**) total phenols; (**b**) total flavonoids; (**c**) water-soluble antioxidant metabolites, expressed as ascorbate equivalents—WS-AOM; (**d**) lipid-soluble antioxidant metabolites expressed as α-tocopherols equivalents—LS-AOM; (**e**) DPPH radical scavenging activity; (**f**) ferric reducing antioxidant power—FRAP. Values are given as mean ± SE (*n* = 4); different letters indicate significant differences assessed by the Fisher LSD test (*p* ≤ 0.05) after performing multifactor ANOVA analysis.

## Data Availability

All data are contained within the article.

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
