# Peer review of "Different Responses to Water Deficit of Two Common Winter Wheat Varieties: Physiological and Biochemical Characteristics"

_plants, 2023, doi:10.3390/plants12122239_

Round 1

Reviewer 1 Report

The actuality of the research presented in the manuscript is clear, given the impacts of climate change experienced and predicted so far. The wheat is the most important cereal crop for most of the countries, so monitoring its response to drought provides important knowledge.

The research presented in the manuscript gives a good overview of the reactions of two different wheat cultivars from a plant physiological and biochemical point of view. The quantity and quality of the results presented are suitable for publication.

However, I propose a structural change to make the manuscript clearer and more compact. Although the Plants journal allows authors to choose whether to prepare a separate Discussion chapter or to discuss the results directly when presenting them, in this case it is disadvantageous to separate the two sections. In many places in the Discussion chapter, the authors repeat their own results, referring to figures again, which unnecessarily increases the length of the manuscript and makes it difficult to read. (Two samples: Rows 378-380, 459-462.) Moreover, the text in Rows 363-368 is again a summary of the description of the experiment. By combining the two chapters, repetition could be avoided and the importance of the analysis of the parameters could be included in the Discussion chapter as an "introduction" to each parameter.

If this combination does not take place, I suggest that the Discussion chapter follows the structure of the Results chapter. Or, if several parameters discussed in separate subsections of the Results are merged in the Discussion, this should be indicated in the subheadings there (e.g. in case of RWC, EL; or in case of thermoluminescence).

In addition to the structural change, I would like to comment/suggest the following:

The manuscript presents a short experiment with young, three-leaved plants. More literature is needed to demonstrate how useful they are. How indicative they are of stress tolerance/susceptibility of later developmental stages, and how often wheat generally encounters drought at this stage. (Or rather, is wheat already at a later developmental stage, when most growing areas are more likely to have drought). All this should perhaps be clarified in the Introduction chapter, present more strongly the importance of the research topic.

All we know about the extent of the drought stress applied is that the plants were not watered for three or seven days. It was not demonstrated how the water content of the root medium changed/decreased, so how much stress actually induced the changes in the plant.

It should be clearly stated how many plants were grown per pot (1?).

It is not clear how many times the control plants were sampled. According to Rows 592-594, maybe three times, as well as on "treated" plants. The correct experimental methodology requires this, since, although there were only a few days' difference between samples, they represented a relatively large part of the plants' life (21-day-old plants!) It is also necessary to specify on which day the stress treatment (not watering) started.

In which part of leaf 3 were the fluorescence studies performed and from which part of leaf 3 were the thylakoid membranes isolated?

In the case of Figure 3, it is disturbing that the length of the y-axis titles of the two graphs below each other makes the two axis titles almost overlap. It would be sufficient to write it once (together).

AFTER MAKING THE PROPOSED EDITORIAL CHANGES AND THE REQUESTED ADDITIONS, I BELIEVE THAT THE MANUSCRIPT MAY BE OF A SUITABLE STANDARD FOR PUBLICATION IN THE JOURNAL PLANTS.

Author Response

attached file

Reviewer 2 Report

The study is interesting because authors examined the responses of two hybrid wheat varieties (Gizda and Fermer) with different drought resistance to moderate (3 days) and severe (7 days) drought stress, as well as their post-stress recovery to understand their underlying defense strategies and adaptive mechanisms in more detail.

Manuscript written well, idea and objectives are clear, however here are few suggestion for further improvement of manuscript.

-'Drought resistance to moderate (3 days) and severe (7 days)' On which basis the days were selected?

-It would be better if authors briefly explain 'Isolation of Thylakoid Membranes' and 'Antioxidant Activity Analysis' procedure in materials and methods.

-Discuss your results by comparing with some more recent research.

-Conclusion is too complicated, rephrase with main/key findings.

overall seems fine, few/minor editing required

Author Response

attached file

Reviewer 3 Report

Dear Authors,

According to the study of the manuscript with the number “plants-2419610" entittled "Different Responses to Water Deficit of Two Common Winter Wheat Varieties: Physiological and Biochemical Characteristics" I did not find any flaws except for some minor correction/improvement so, the manuscript can be accepted in its original form after that. 

For Photosynthetic pigment parameters autors are requested to express their findings in the form of percent (%) increase or decrease as they did for RWC and other parameters, as it give reader more clear idea about the results instead of giving the values. Secondly, their is no need of mentioning the values of SE in result section as it is already clearly given in graphs.

Similar discrepancy was observed in Line 223-224

Line No 155 authors have mentioned "The maximum quantum efficiency of  PSII electron transport, Y(II), decreased by 10% after 3 days of drought...." This % is for which variety?? Kindly clarify.

The overall quality of paper is very good, very well written and very informative.

Kindly incorporate the above mentioned suggestions to improve the quality of your paper

Good Luck

Author Response

attached file

Round 2

Reviewer 1 Report

I thank the authors for completing and explaining the requested information.